# Reactogenicity to major tuberculosis antigens absent in BCG is linked to improved protection against *Mycobacterium tuberculosis*

Nacho Aguilo[1,2,*], Jesus Gonzalo-Asensio[1,2], Samuel Alvarez-Arguedas[1,2], Dessislava Marinova[1,2], Ana Belen Gomez[1,2], Santiago Uranga[1,2], Ralf Spallek[3], Mahavir Singh[3], Regine Audran[4], François Spertini[4] & Carlos Martin[1,2,5,*]

MTBVAC is a live-attenuated *Mycobacterium tuberculosis* vaccine, currently under clinical development, that contains the major antigens ESAT6 and CFP10. These antigens are absent from the current tuberculosis vaccine, BCG. Here we compare the protection induced by BCG and MTBVAC in several mouse strains that naturally express different MHC haplotypes differentially recognizing ESAT6 and CFP10. MTBVAC induces improved protection in C3H mice, the only of the three tested strains reactive to both ESAT6 and CFP10. Deletion of both antigens in MTBVAC reduces its efficacy to BCG levels, supporting a link between greater efficacy and CFP10- and ESAT6-specific reactogenicity. In addition, MTBVAC (but not BCG) triggers a specific response in human vaccinees against ESAT6 and CFP10. Our results warrant further exploration of this response as potential biomarker of protection in MTBVAC clinical trials.

[1] Grupo de Genética de Micobacterias, Dpto. Microbiología, Medicina Preventiva y Salud Pública, Universidad de Zaragoza, C/Domingo Miral s/n, Zaragoza 50009, Spain. [2] CIBER Enfermedades Respiratorias, Instituto de Salud Carlos III, Madrid 28029, Spain. [3] LIONEX GmbH, Salzdahlumer Straße 196, Braunschweig 38126, Germany. [4] Division of Immunology and Allergy, Centre Hospitalier Universitaire Vaudois (CHUV), Lausanne CH-1011, Switzerland. [5] Servicio de Microbiología, Hospital Universitario Miguel Servet, ISS Aragón, Paseo Isabel la Católica 1-3, Zaragoza 50009, Spain. * These authors jointly supervised this work. Correspondence and requests for materials should be addressed to N.A. (email: naguilo@unizar.es) or to C.M. (email: carlos@unizar.es).

Tuberculosis (TB) disease causes around 1.5 million deaths per year and is one of the leading airborne infectious diseases affecting developing countries. The growing threat of antibiotic-resistant strains makes TB treatment difficult or even impossible. Thus, development of new vaccines able to prevent respiratory forms of TB will have a tremendous impact in preventing transmission and control of the disease[1,2].

The current TB vaccine, Bacille Calmette–Guerin (BCG), is a live-attenuated strain of the bovine pathogen *Mycobacterium bovis* described to be protective against severe forms of TB (meningitis and milliary TB) in children, but with inconsistent and variable protection against pulmonary TB, which is responsible for spread of the most common form of the disease in adolescents and adults[3]. Developed a century ago by repeated subculture, the principal genetic basis for BCG attenuation is the loss of RD1 region, that includes the genes codifying for the major virulence factors ESAT6 and CFP10, and other ones involved in ESAT6/CFP10 secretion[4,5]. In addition to being major virulence determinants, ESAT6 and CFP10 are the most immunogenic proteins of *M. tuberculosis*, and ESAT6 is included in the construction of promising TB vaccine candidates[6–8].

MTBVAC is a live vaccine candidate rationally attenuated from the clinical isolate *M. tuberculosis* Mt103 (ref. 9), which belongs to lineage 4 (Euro–African–American), one of the most widespread lineages of *M. tuberculosis* that conserves most of the T-cell epitopes described for TB[10]. MTBVAC attenuation is conferred by two independent unmarked deletions in the *phoP* and *fadD26* virulence genes, in accordance with the Geneva Consensus requirements for the construction of live-attenuated mycobacterial vaccines[11]. PhoP is a transcription factor that controls ∼2% of the coding capacity of *M. tuberculosis* genome, including production of immunomodulatory cell-wall lipids and secretion of ESAT6 so that *phoP* mutants produce ESAT6 but are unable to export it[12]. Deletion of *fadD26* leads to complete abolishment of phtioceroldimycocerosates (PDIM) synthesis, known to be virulence lipids constituent of the envelope[13]. MTBVAC has been characterized in different animal models showing a safe, immunogenic and protective profile[9,14–16]. MTBVAC is so far the first and only live-attenuated *M. tuberculosis* vaccine that has entered clinical development. In 2015, MTBVAC successfully completed the first-in-human phase 1 clinical trial for safety and immunogenicity in healthy adult volunteers in Switzerland[17], which has allowed it to reach clinical evaluation in newborns (clinical trial identifier: NCT02729571) and adolescents (NCT02933281) in TB-endemic countries. MTBVAC is being developed as preventive strategy against all forms of the disease in newborns and adults.

Even though MTBVAC has demonstrated improved protection compared to BCG in adult and newborn animal models[9,14,16], the mechanisms underlying its protective efficacy have not been characterized. Elucidating these mechanisms remain crucial for the identification of vaccine-specific biomarkers, which would accelerate the clinical development of MTBVAC and would help anticipate results from other TB vaccine candidates in the preclinical and clinical pipeline.

The main support for the hypothesis that MTBVAC may confer improved efficacy relative to BCG in humans is that the MTBVAC genome maintains the whole T-cell antigen repertoire of the human pathogen *M. tuberculosis*. This repertoire includes those antigens located in RD1 and consequently absent in BCG[5]. Here we assess the protection mediated by MTBVAC in three different mouse genetic backgrounds, selected for their differential ability to recognize and present ESAT6- and CFP10-derived epitopes, to evaluate the potential protective role of the immunogenicity mediated by these two major antigens present in MTBVAC. Our results suggest that MTBVAC immunogenicity against these major antigens, absent in BCG, contributes to vaccine-induced protection, and highlight the importance of host genetics in protective efficacy.

## Results

### ESAT6/CFP10 reactogenicity depends on host genetic background.
Unlike BCG, MTBVAC contains the RD1 region and as expected, we found that it differentially expressed *esat6* (*esxA*-Rv3875) and *cfp10* (*esxB*-Rv3874) (Fig. 1a). Accordingly, ESAT6 and CFP10 proteins were present in the intracellular fraction of MTBVAC but not BCG (Fig. 1b). Analysis of the secreted protein fraction of MTBVAC revealed that ESAT6 was not released by MTBVAC, which is in agreement with our previous results[9]. Conversely, we found CFP10 in the MTBVAC extracellular fraction (Fig. 1b,c), which was unexpected as these two proteins are described to be co-secreted[18]. We next sought to evaluate ESAT6- and CFP10-specific reactogenicity in different mouse genetic backgrounds following MTBVAC vaccination. To do that, we subcutaneously immunized with MTBVAC C57BL/6, BALB/c or C3H/HeNRj mouse strains, each expressing a different haplotype of the major histocompatibility complex (MHC); H-2b, H-2d and H-2k, respectively. After vaccination, splenocytes were stimulated with either purified protein derivative (PPD) or single antigens ESAT6, CFP10 or Ag85B, with the objective to elucidate the possible influence of the MHC haplotype in the specific response against these proteins. Our data revealed substantial differences among the three different mouse strains (Fig. 2). MTBVAC vaccination conferred immunogenicity to ESAT6 and Ag85B in C57BL/6 (Fig. 2a); to none of the single antigens tested in this work in the BALB/c background (Fig. 2b); and to ESAT6 and CFP10, but not to Ag85B, in the C3H/HeNRj mouse strain (Fig. 2c). PPD-positive IFNγ response in the BALB/c mice suggested recognition by the H-2d haplotype of other undefined exported mycobacterial proteins different from the single antigens studied here, but present in the PPD. We found that BCG did not induce any response to ESAT6 or CFP10 stimulation irrespective of the MHC alleles. Our results are in accordance with previous work showing the exclusive capacity of the H-2k haplotype of the C3H mouse strain to recognize CFP10-derived peptides, in comparison to H-2b or H-2d of C57BL/6 and BALB/c strains, respectively[19].

Flow cytometry analysis of CD4+IFNγ+ splenocytes confirmed the IFNγ secretion profile observed by ELISA (Supplementary Fig. 1a–c). In addition, analysis of IFNγ-positive splenocytes indicated that around 70% of this population corresponded to CD4+ cells, whereas <5% were CD8+ cell, suggesting that CD4+ T cells are major contributors of IFNγ production on antigen stimulation *ex vivo* (Supplementary Fig. 1d,e).

Since our data revealed lack of Ag85B-specific response in BCG-vaccinated C57BL/6 mice (Fig. 2a), we compared *fbpB* gene expression and Ag85B secretion between BCG and MTBVAC. Even though *fbpB* gene was similarly expressed by both vaccines (Fig. 1a), Ag85B protein was not detected in the secreted fraction of BCG (Supplementary Fig. 2), which could explain our *in vivo* observations. Absence of Ag85B in the secreted fraction of BCG could be justified by a recently described Phe140Leu polymorphism in the *fbpB* gene of all BCG substrains predicting an unstable protein[20].

### CFP10 and ESAT6 reactogenicity contributes to protection.
Eight weeks following BCG or MTBVAC subcutaneous vaccination, mice from the three strains were intranasally infected with a low-dose challenge of *M. tuberculosis* H37Rv (infectivity 20 CFU). Four weeks later, bacterial load in lungs (Fig. 2d–f) and spleen

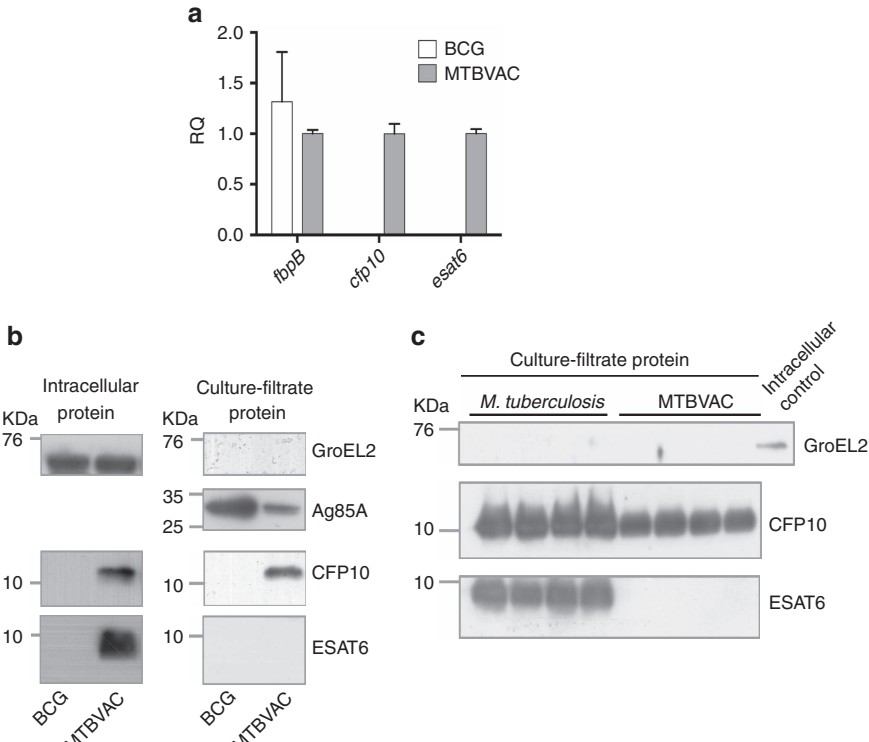

**Figure 1 | Expression and secretion of ESAT6 and CFP10 by MTBVAC. (a)** Normalized expression (using *sigA* as housekeeping gene) of *fbpB*, *esat6* and *cfp10* genes in log-phase broth-cultured BCG and MTBVAC. Data in the graph are represented as the relative quantity (RQ) using MTBVAC as comparator. Results are the average from triplicate experiments. Data are mean ± s.d. **(b,c)** Immunoblot analysis of GroEL2, Ag85A, ESAT6 and CFP10 in BCG, MTBVAC and *M. tuberculosis* protein extracts. **(b)** Intracellular (left) and supernatant (right) fractions of BCG and MTBVAC were assessed. **(c)** Comparison between *M. tuberculosis* and MTBVAC-secreted fractions. A BCG cell lysate sample was used as positive control for GroEL2 detection. Four independent protein extractions of each strain were included in the analysis. Full blots are shown in Supplementary Fig. 7a–c.

(Fig. 2g–i) revealed protection of both vaccines as assessed by bacterial load reduction relative to unvaccinated controls. Nevertheless, whereas both BCG and MTBVAC protected to a similar extent in C57BL/6 (Fig. 2d,g) and BALB/c (Fig. 2e,h) mice, we observed a more pronounced bacterial load reduction in C3H mice vaccinated with MTBVAC, as compared to BCG (Fig. 2f,i).

As an additional control, we studied the replication and dissemination profile of MTBVAC and BCG in the draining lymph nodes and spleen of C3H mice (Supplementary Fig. 3). The pattern observed resulted comparable, with no profound differences between both vaccines, similar to that published previously in BALB/c mice[9]. Altogether, these results suggest that differences in protection between BCG and MTBVAC in C3H mice were not due to a different biological behaviour of the two vaccines.

To evaluate the contribution to protection of MTBVAC-induced immune responses against CFP10 and ESAT6, we constructed an MTBVAC substrain with deleted *cfp10* and *esat6* genes (MTBVACΔE6C10) (Fig. 3a). To construct this mutant, we used a BAC-rec strategy that permitted us to substantially accelerate the generation of deletion mutants in *M. tuberculosis* (Supplementary Fig. 4). We found that mice vaccinated with MTBVACΔE6C10 showed inability to elicit specific responses to CFP10 and ESAT6 (Fig. 3b,c). Interestingly, IFNγ production in C3H mice following PPD stimulation was lower in the MTBVACΔE6C10-vaccinated group compared to MTBVAC-vaccinated control, suggesting that ESAT6 and CFP10 are likely the main immunodominant antigens in this genetic background. Importantly, MTBVACΔE6C10 triggered an Ag85B-specific response equivalent to the parental strain (Fig. 3b), indicating that genetic manipulation did not alter the ability of MTBVAC knockout to elicit antigen-specific immunity.

Protective efficacy studies in C57BL/6 mice (Fig. 3d,g), reactive to ESAT6 but not CFP10, showed that immunization with MTBVACΔE6C10 or parental MTBVAC-conferred similar protection, suggesting that ESAT6-specific immune response is not sufficient for optimal efficacy of MTBVAC. Remarkably, MTBVACΔE6C10 vaccination of C3H mice, the only genetic background reactive to both antigens studied in the present work, conferred lower protection than MTBVAC vaccination and comparable to BCG (Fig. 3f,i).

Confirming the capacity of I-A$^k$, I-E$^k$ molecules to present CFP10-derived peptides to CD4+ cells, we observed *in vitro* that CD4+ cells purified from C3H mice vaccinated with MTBVAC, but not with MTBVACΔE6C10, produced IFNγ when incubated with CFP10-pulsed syngeneic bone-marrow-derived macrophages in an antigen concentration-dependent fashion (Supplementary Fig. 5).

In addition, an MTBVAC knockout for Ag85B was constructed to evaluate the role of this antigen in MTBVAC-conferred protection. Results showed that in the absence of Ag85B-specific response, MTBVAC efficacy remained unaffected in any of the three genetic backgrounds tested (Fig. 4).

**Increased *cfp10* and *esat6* gene expression *in vivo*.** Even though we observed that MTBVAC induced a strong response specific for CFP10, ESAT6 and Ag85B, our results indicate that only reactogenicity against RD1-containing antigens, but not Ag85B,

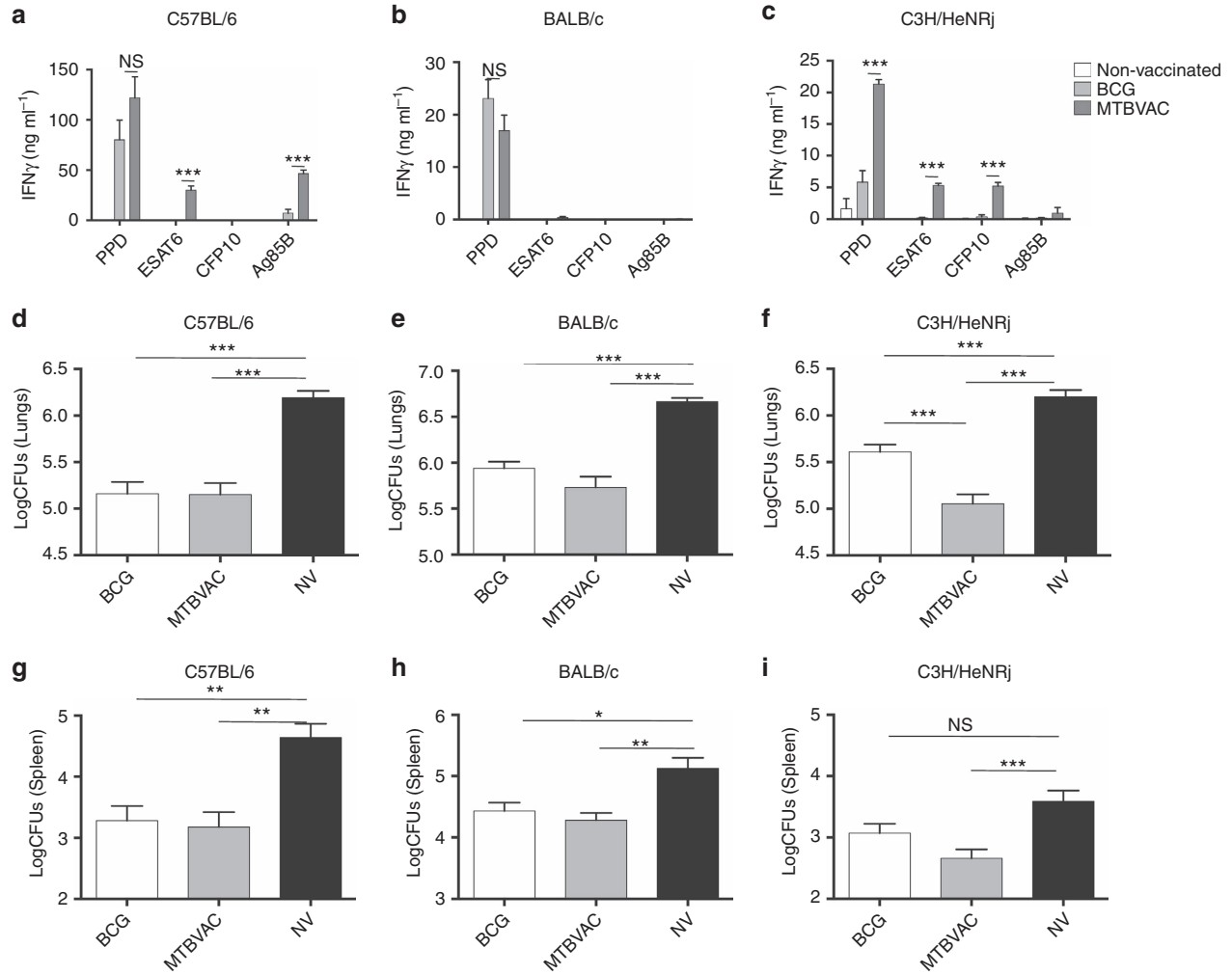

**Figure 2 | Improved protection of MTBVAC compared to BCG is dependent on the host genetics.** (**a,b,c**) Antigen-specific IFNγ production following stimulation with PPD (5 μg ml$^{-1}$), ESAT6 (2 μg ml$^{-1}$), CFP10 (2 μg ml$^{-1}$) and Ag85B (2 μg ml$^{-1}$) during 48 h of splenocytes from mock-, BCG- and MTBVAC-vaccinated C57BL/6, BALB/c and C3H/HeNRj mice. (**d–i**) Lung (**d,e,f**) and spleen (**g,h,i**) bacterial load 4 weeks post low-dose H37Rv intranasal challenge. C57BL/6 (**d,g**), BALB/c (**e,h**) and C3H/HeNRj (**f,i**) were vaccinated with BCG, MTBVAC or unvaccinated eight weeks before challenge. (**a,b,c**) Data are representative from one of two independent experiments (n = 5 mice per group per experiment). (**d–i**) Data in the graphs represent a pool of two independent experiments (n = 12 mice per group). All data are mean ± s.e.m. (**a,b,c**) NS, non-significant; ***P < 0.001 by unpaired t-student test. (**d–i**) NS, non-significant; *P < 0.05; **P < 0.01; ***P < 0.001 by one-way ANOVA and Bonferroni post-test.

contributed to protection. We hypothesized that this difference could be related to the expression level of each of these antigens by *M. tuberculosis* on early lung infection. Indeed, it has been described that *M. tuberculosis* downregulates *fbpB* (Ag85B-codifying gene) expression *in vivo*, as a mechanism to impair host recognition[21]. Thus, we compared *in vivo* expression of *fbpA* and *fbpB* with *esat6*, *cfp10*, *espA* and *espC* genes. EspA and EspC are essential factors for ESAT6 and CFP10 secretion[22–24]. Thus, we intranasally infected C3H (Fig. 5a) and C57BL/6 (Fig. 5b) mice with H37Rv, and 4 weeks later we isolated intrapulmonary bacteria and analysed gene expression by RT–qPCR. Expression levels were compared under *in vivo* conditions and in culture using standard 7H9 media. Our data revealed a comparable expression level for all the genes studied under *in vitro* culture. Conversely, when studied *in vivo*, we detected in both mouse strains an enriched expression of *esat6* and *cfp10* genes, as well as *espA* and *espC*, relative to *fbpA* and *fbpB*.

Of note, higher *in vivo* expression of *esat6* relative to *fbpB* has been previously described[6,25]. Our results extend these

observations to other genes from the ESX-1 secretion system, including *cfp10* and those involved in the secretion of ESAT6.

**MTBVAC induces a CFP10-specific response in human vaccinees.** As our data showed a link between CFP10-/ESAT6-induced reactogenicity following MTBVAC immunization and improved protection compared to BCG, we sought to evaluate the induction of responses against these antigens in MTBVAC-vaccinated humans. We used the ESAT6- and CFP10-specific ELISPOT data from the MTBVAC Phase 1 trial performed in adults in Switzerland, which had concluded that none of the MTBVAC vaccinees exceeded the spot threshold for positive latent *M. tuberculosis* infection at the end of active follow-up study[17]. We thus compared results from each individual before (day 0) and after (day 210) vaccination with MTBVAC or BCG (5 × 10$^5$ CFU vaccination dose), finding a significant increase in the CFP10-specific response in the MTBVAC vaccinees (Fig. 6a). For ESAT6, though results showed a rising tendency in the group of MTBVAC, the increment was not significant (Fig. 6c).

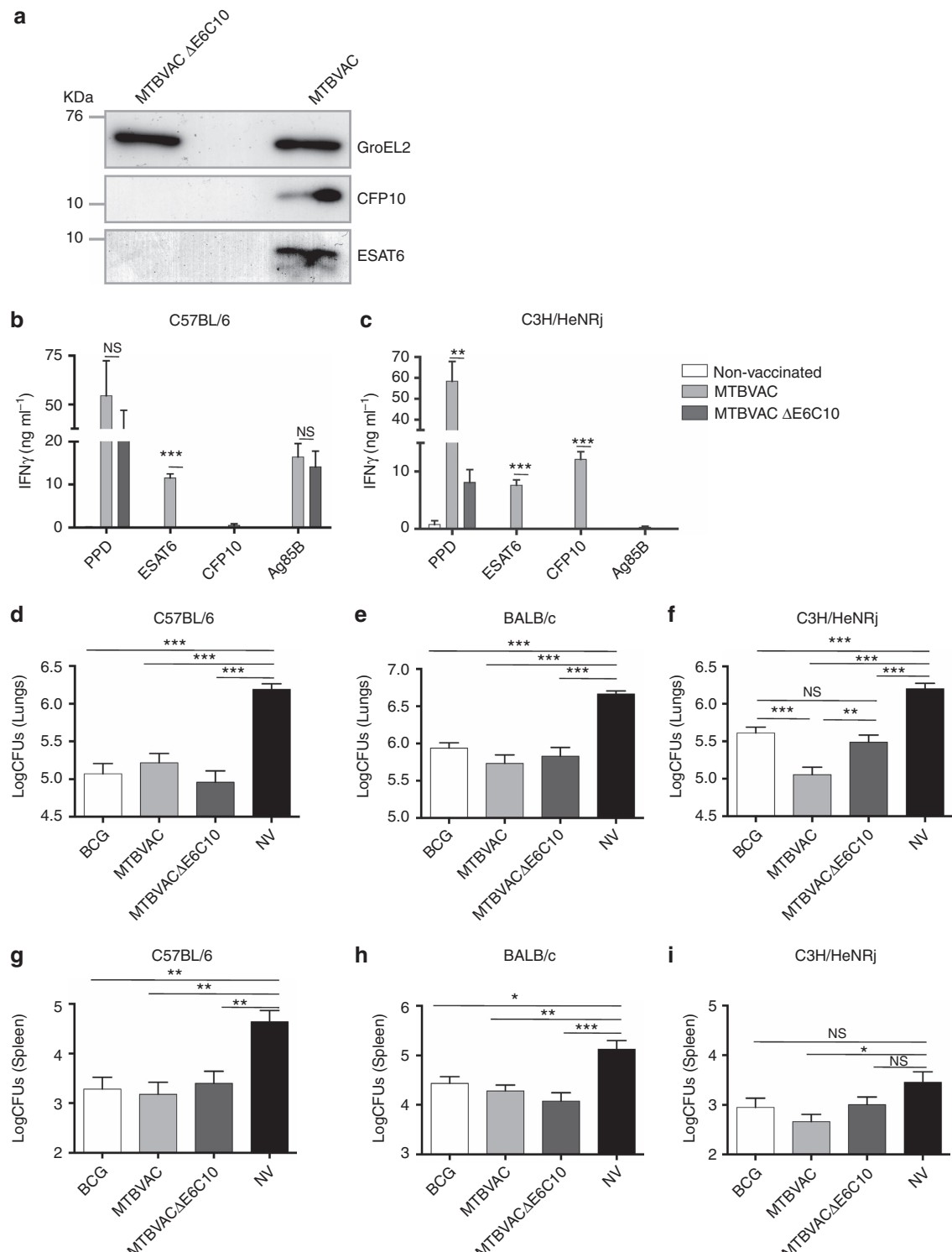

**Figure 3 | MTBVAC-induced immune response specific to ESAT6 and CFP10 is protective.** (**a**) Immunoblot analysis of GroEL2, ESAT6 and CFP10 in MTBVAC and MTBVACΔE6C10 lysate samples. Full blot is shown in Supplementary Fig. 7d. (**b,c**) Antigen-specific IFNγ production following stimulation with PPD (5 μg ml$^{-1}$), ESAT6 (2 μg ml$^{-1}$), CFP10 (2 μg ml$^{-1}$) and Ag85B (2 μg ml$^{-1}$) during 48 h of splenocytes from mock-, MTBVAC- and MTBVACΔE6C10-vaccinated C57BL/6 (left) and C3H/HeNRj (right) mice. (**d–i**) Lung (**d,e,f**) and spleen (**g,h,i**) bacterial load 4 weeks post low-dose H37Rv intranasal challenge. C57BL/6 (**d,g**), BALB/c (**e,h**) and C3H/HeNRj (**f,i**) were vaccinated with BCG, MTBVAC, MTBVACΔE6C10 or unvaccinated 8 weeks before challenge. (**b,c**) Data are representative from one of two independent experiments ($n = 5$ mice per group per experiment). (**d,e,g,h**) Data are derived from $n = 6$. (**f,i**) Data represent a pool of two independent experiments ($n = 12$ mice per group). All data are mean ± s.e.m. (**b,c**) NS, non-significant; **$P < 0.01$; ***$P < 0.001$ by unpaired $t$-student test. (**d–i**) NS, non-significant; *$P < 0.05$; **$P < 0.01$; ***$P < 0.001$ by one-way ANOVA and *Bonferroni* post-test.

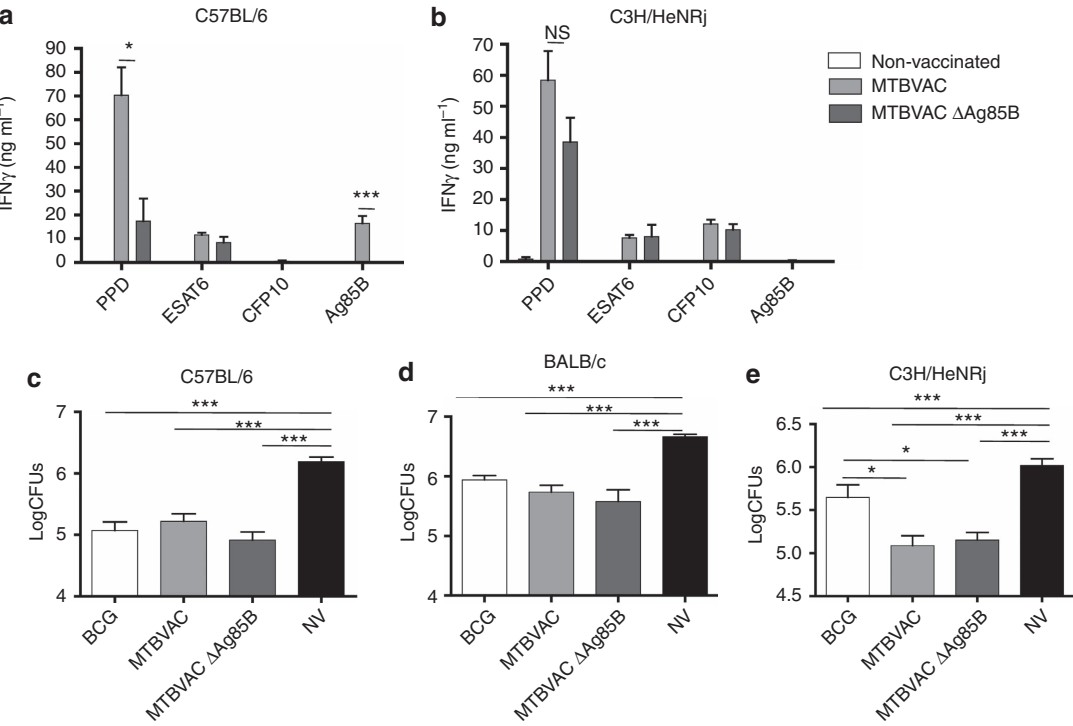

**Figure 4 | Ag85B-specific immunogenicity induced by MTBVAC is not protective.** (a,b) Antigen-specific IFNγ production following stimulation with PPD (5 µg ml), ESAT6 (2 µg ml$^{-1}$), CFP10 (2 µg ml$^{-1}$) and Ag85B (2 µg ml$^{-1}$) during 48 h of splenocytes from mock-, MTBVAC- and MTBVACΔAg85B-vaccinated C57BL/6 (left) and C3H/HeNRj (right) mice. (c–e) Lung bacterial load 4 weeks post low-dose H37Rv intranasal challenge. C57BL/6 (c), BALB/c (d) and C3H/HeNRj (e) were vaccinated with BCG, MTBVAC, MTBVACΔAg85B or unvaccinated 8 weeks before challenge. (a,b) Data are representative from one of two independent experiments (n = 5 mice per group per experiment). (c–e) Data are derived from n = 6. All data are mean ± s.e.m. (a,b) NS, non-significant; **P < 0.01; ***P < 0.001 by unpaired t-student test. (c–e) *P < 0.05; **P < 0.01; ***P < 0.001 by one-way ANOVA and Bonferroni post-test.

Confirming assay specificity, none of the BCG-vaccinated participants showed any increment in CFP10- and ESAT6-specific responses at the end of the study (Fig. 6b,d). Therefore, MTBVAC induces a CFP10-specific immune response in humans.

## Discussion

The MHC is composed of several highly polymorphic loci resulting in a wide interindividual variability of MHC haplotypes, which can have great influence in host-specific responsiveness to vaccines, as individual haplotype may determine antigen immunodominance associated with protection[26]. To date, there are no defined BCG-specific immunodominant antigens associated with efficacy against TB. The identification of such correlates of protection for new TB vaccines, as MTBVAC, would greatly accelerate clinical development to efficacy trials.

A plausible explanation for the poor protection of BCG against pulmonary TB is the loss of several genomic regions, including RD1 (refs 5,8), which encodes ESAT6, CFP10 and part of their secretory machinery contained in the ESX-1 secretion system. Despite their low molecular weight, ESAT6 and CFP10 are the top two antigens with the highest proportion of peptides recognized by human MHC haplotypes[20,27]. Importantly, loss of RD1 has also been reported as the main basis for attenuation of BCG[5]. Our results indicated that, unlike other major antigens not related with virulence, such as those belonging to the Ag85 complex, ESAT6, CFP10 and the machinery involved in their secretion are highly over expressed on early infection, which is probably the result of their role in virulence and infection establishment. Indeed, in agreement with these observations, the

phagosomal escape of *M. tuberculosis*, which is mediated by ESAT6, occurs as early as day 3 after *in vitro* macrophage infection[28]. Thus, a vaccine inducing an effective response against ESAT6 and CFP10, such as MTBVAC, may have the advantage to target antigens highly expressed during early *M. tuberculosis* infection. This would probably lead to a better recognition of *M. tuberculosis*-infected cells, whose MHC molecules should be coated with epitopes derived from dominant antigens. Conversely, vaccination-induced immunogenicity against other antigens, such as Ag85A or Ag85B, underexpressed and therefore less represented in the antigen repertoire at early infection stages, would be less efficient against initial infection as a consequence of a poorer recognition of pathogen-infected cells. In line with this, other authors have demonstrated downregulation of Ag85B expression during early *M. tuberculosis* infection[21].

Our efficacy data in mice seem to contrast with previous results using a recombinant BCG vaccine overexpressing Ag85B (rBCG30), which showed an improved protection relative to BCG in an outbred guinea pig model[29]. However, the different animal model used in that study (more sensitive to TB) and the different readout utilized to measure vaccine efficacy (survival) could account for such differences.

An important proportion of human MHC class II haplotypes recognizes ESAT6- and CFP10-derived peptides[30]. This could be explained by the important roles played by both proteins in virulence, eliciting an evolutionary pressure leading to the selection of MHC human haplotypes that recognize highly conserved peptides derived from these antigens. Our results suggest that the immunodominance of ESAT6 and CFP10 in humans is not reflected in mice, as only one of the three haplotypes tested simultaneously presents both antigens.

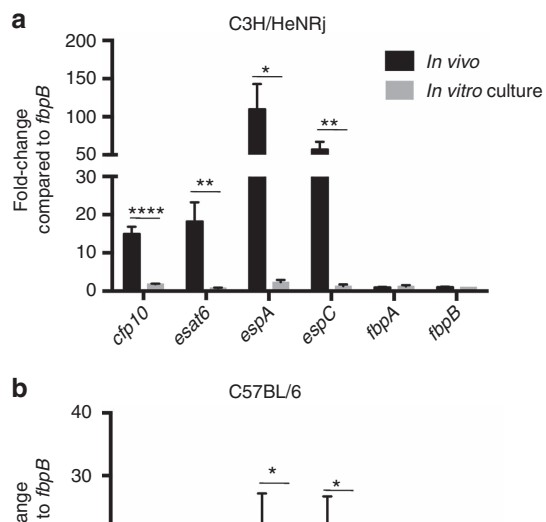

**Figure 5 | Enriched *in vivo* expression of genes from ESX-1 secretion system.** (**a,b**) Expression of *esat6*, *cfp10*, *espA*, *espC*, *fbpA* and *fbpB* genes (normalized with *16s* gene expression) from H37Rv isolated from lungs from C3H/HeNRj (**a**) and C57BL/6 (**b**) 4 weeks after high-dose ($10^3$ CFU) intranasal challenge, in comparison to expression obtained under 7H9-culture *in vitro* conditions. Normalized results are represented for each gene and experimental condition as the fold-change induction in comparison to *fbpB* expression level. *In vivo* data are derived from $n = 4$ mice (**a**) and $n = 6$ mice (**b**). *In vitro* data represent a pool of four independent RNA extractions. All data are mean ± s.e.m. *$P < 0.05$; **$P < 0.01$; ***$P < 0.001$; ****$P < 0.0001$ by unpaired *t*-student test.

This difference could be due to lack of co-evolution of the mouse MHC loci with ESAT6- and CFP10-expressing mycobacteria. Indeed, *Mycobacterium microti*, the mycobacterial species that naturally infects rodents, lacks the RD1$^{mic}$ region that includes both genes[5]. Conversely, those species of the *M. tuberculosis* complex able to infect humans (*M. canettii*, *M. africanum* and *M. tuberculosis*) have retained intact *esat6* and *cfp10* genes as well as their secretory machinery[31,32].

Our previously published results in MTBVAC-vaccinated newborn C57BL/6 mice showed a slight but significant improvement of MTBVAC-conferred protection relative to BCG in lungs of infected pups with an aerosol challenge dose of H37Rv similar to the one used in the present work[16]. This previous work also demonstrated that, unlike the comparable immunogenicity of MTBVAC and BCG observed in adult mice, MTBVAC was much more immunogenic (following PPD stimulation) than BCG when animals were vaccinated at birth, which could account for such difference in protection. In two other studies[9,33], where we vaccinated C57BL/6 adult mice, we also found an improved protection in the MTBVAC group, although in those works we used a much higher initial H37Rv challenge dose (100 and 1,000 CFU) than the one we used in the present work (20 CFU). In line with this, previous data in a guinea pig model revealed that MTBVAC only improved BCG protection in high-dose challenge experiments[14], suggesting the contribution of undefined

mechanism(s) of protection occurring at different experimental conditions than the tested in this work.

Our data indicate that both H-2b (C57BL/6) and H-2k (C3H) haplotypes recognized ESAT6-derived epitopes following MTBVAC immunization. However, MTBVAC only improves BCG-conferred protection in C3H mice, the only of the three tested genetic backgrounds whose MHC haplotype presents CFP10-derived peptides. This result could suggest a particular contribution of CFP10-specific reactogenicity to MTBVAC protection. Interestingly, we found that MTBVAC secretes CFP10 but not ESAT6. Provided secreted proteins are differentially processed by the host immune system in comparison to intracellular proteins[34], we speculate that ESAT6 and CFP10 might induce different responses in MTBVAC vaccinees, which might lead to differential protection. In this regard, it could be relevant in the future to construct MTBVAC mutant strains for ESAT6 and CFP10 separately, to study these potential mechanisms in more detail.

It has been reported that both ESAT6 and CFP10 are co-secreted by *M. tuberculosis*[18]. Therefore, further work is needed to elucidate the molecular mechanisms behind the CFP10-independent secretion of ESAT6 in MTBVAC. Two previous publications reported that uncoupled ESAT6 and CFP10 secretion can occur under some conditions[35,36]. In these studies, authors found that CFP10 is secreted independently of ESAT6 in the presence of an aberrant ESX-1 system. Thus, we speculate that since PhoP is a regulator of different genes encoding ESX-1 components[12,37,38], MTBVAC may express a dysfunctional ESX-1 system responsible of the observed phenotype.

Our study underlines the role of host genetics in vaccine-induced protection and how genetic background determines antigen immunodominance, as shown with CFP10 in the C3H mouse strain. In line with this finding, the importance of host genotype for tuberculosis susceptibility and vaccine protection in the mouse model has been recently reported[39]. Thus, our results indicate that MTBVAC-vaccinated mice reactogenic for CFP10 and ESAT6 are more protected against tuberculosis than non-reactogenic strains. In humans, responders to these antigens comprise around 60–80% of individuals[30,40]. We speculate that a vaccine expressing ESAT6 and CFP10 might confer improved protection in these individuals (in comparison with BCG); however, further clinical data would be needed to test this hypothesis.

One of the handicaps of a vaccine whose protective mechanism is based on ESAT6- and/or CFP10 antigens remains in their potential interference with the latent TB diagnostic test. Data provided by the first-in-human clinical trial of MTBVAC in adults indicated that the ELISPOT response elicited by MTBVAC following ESAT6 or CFP10 stimulation was below the cutoff established for tuberculosis infection[17], suggesting no interference of MTBVAC at least in this target population. Interestingly, it has been reported that individuals with a quantiFERON value greater than $4\,\mathrm{IU\,ml^{-1}}$ have a higher probability of developing active TB, while individuals with a value between 0.35 and 4 (cutoff for positivity is 0.35) have a similar probability than negative individuals[41]. Thus, this study opens the possibility to reconsider the cutoff of this technique to identify individuals with a high risk of developing active TB. Current clinical trials in TB-endemic countries in neonates (NCT02729571) are further evaluating quantiFERON conversion following MTBVAC vaccination to evaluate this question in more detail.

MTBVAC is the only vaccine in the TB vaccine pipeline of candidates in clinical evaluation that primes both CFP10- and ESAT6-specific immune responses[42]. Our data suggest that this strategy might be effective in protecting from pulmonary TB,

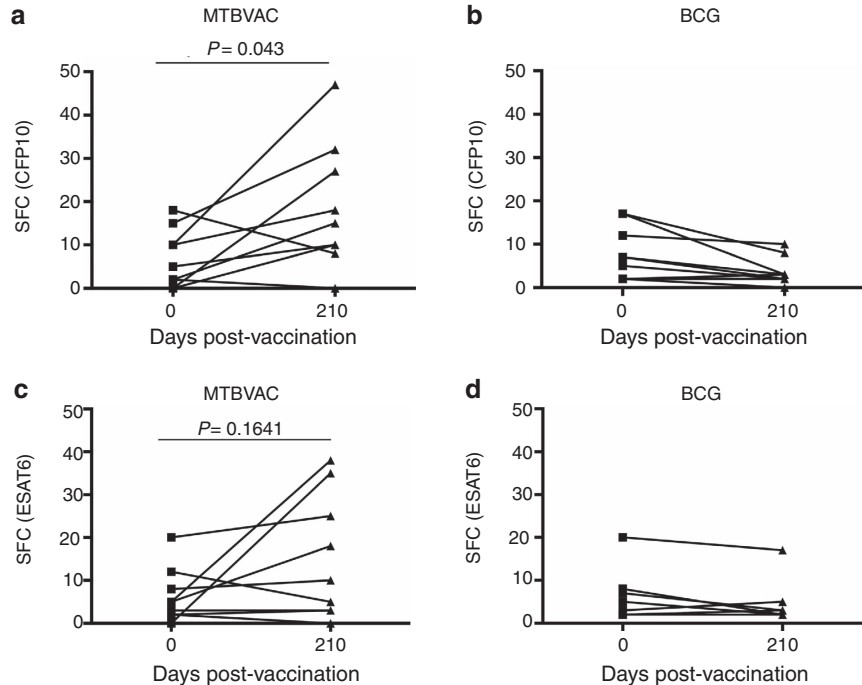

**Figure 6 | Vaccination with MTBVAC induces a CFP10-specific immune response in humans.** Specific IFNγ ELISPOT results from the first-in-human MTBVAC clinical trial are shown for CFP10 (**a,b**) and ESAT6 (**c,d**), comparing for each volunteer the number of spots pre- and post-BCG (**b,d**) and MTBVAC (**a,c**) vaccination with $5 \times 10^5$ CFU. Wilcoxon matched-pairs signed rank test was used to compare pre- and post-vaccination status for each vaccine and antigen. *P* value is indicated for MTBVAC vaccinees.

which would have an evident impact on TB transmission. Supporting this hypothesis, prospective cohort studies of persons exposed to individuals with active TB have indicated that latently infected individuals (reactive to CFP10 and ESAT6 stimulation) are more protected against reinfection than non-infected people[43]. Our data support the potential usefulness of the analysis of CFP10- and ESAT6-positive cells after vaccination in future clinical efficacy trials, in the search for a possible biomarker of vaccine-induced protection.

## Methods

**Bacteria.** *M. bovis* BCG Danish (Statens Serum Institute), *M. tuberculosis* MTBVAC (University of Zaragoza) and *M. tuberculosis* H37Rv (Institut Pasteur Paris) strains were grown at 37 °C in Middlebrook 7H9 broth (Difco) supplemented with ADC 10% (Difco) and 0.05% (v/v) Tween-80 (Sigma) or on solid Middlebrook 7H11 (Difco) supplemented with ADC 10%. Bacterial suspensions for vaccination or infection were prepared in PBS from glycerol stocks previously quantified by plating serial dilutions.

**Mouse experiments.** All mice were kept under controlled conditions and observed for any sign of disease. Experimental work was conducted in agreement with European and national directives for protection of experimental animals and with approval from the Ethics Committee from University of Zaragoza (approved protocol PI46/14). No randomization specific methodology was applied to this study.

Female, 8–10 weeks old C57BL/6, BALB/c and C3H/HeNRj mice (Janvier Biolabs) were vaccinated subcutaneously (100 μl) with $10^6$ CFU of vaccine strains in PBS. Eight weeks post vaccination, mice were intranasally challenged with 150 CFU of H37Rv in 40 μl of PBS. Bacterial burden was assessed 4 weeks post challenge by plating homogenized lungs and spleen on solid medium. A group of infected mice was killed 1 day after challenge to determine the initial bacterial load in lungs, which resulted to be ∼ 20 CFU in all experiments (Supplementary Fig. 6).

For immunogenicity studies, mice were vaccinated subcutaneously with $10^6$ CFU of vaccine strains in PBS, and 8 weeks later animals were killed and splenocytes collected. $10^6$ cells were stimulated in 96-well U bottom plates with PPD (Statens Serum Institute, SSI) 5 μg ml$^{-1}$, or 2 μg ml$^{-1}$ of ESAT6, CFP10 or Ag85B (LIONEX GmbH, Braunschweig, Germany) during 48 h for supernatant collection and cytokine detection by ELISA. Cytokine concentration in supernatants was determined with an IFNγ-specific ELISA kit (MabTECH). For

intracellular staining, splenocytes were incubated with antigens for 24 h, and 10 μg ml$^{-1}$ Brefeldin A (Sigma) was added during the last 6 h of incubation. For surface staining, cells were labelled with anti-CD4-FITC (553047, BD Biosciences) and anti-CD3-PerCPVio700 (130-109-883, Miltenyi Biotec), diluted 1:500 and 1:50, respectively, in culture medium with 10% FCS. Then, cells were fixed and permeabilized with the Cytofix/Cytoperm Fixation/Permeabilization Kit (BD Biosciences) following the manufacturer's instructions, and stained with anti-IFNγ-APC (554413, BD Biosciences), diluted 1:200 in permeabilization buffer. Cells were acquired with a Gallios Flow Cytometer (Beckman).

**Construction of MTBVAC mutant substrains.** We developed a BAC-rec strategy to accelerate knockout construction in *M. tuberculosis*, which allowed to generate the desired mutant strains in ∼1 month. To construct the allelic exchange substrate (AES), we used a *M. tuberculosis* H37Rv bacterial artificial chromosome (BAC) library constructed in the pBeloBAC11 vector and introduced into *Escherichia coli* DH10B[44] (a kind gift from Roland Brosch, Institut Pasteur Paris, France). The *E. coli* clone carrying the BAC containing the target gene was identified (Rv221 for *fbpB* and Rv414 for *cfp10-esat6*). Bacteria were made electrocompetent and transformed with pKD46 carrying the red recombinase from lambda phage[45]. The resulting *E. coli* strain was transformed with a PCR product, which contains the kanamycin-resistant marker (Km$^r$) from pKD4 (ref. 45) flanked by 40 bp homology arms to the target gene located in the BAC (using E6C10 P1/E6C10 P2 or Ag85B P1/Ag85B P2 primer pairs, Table 1). After induction of the lambda red recombinase by incubation in the presence of 1 mM L-arabinose, recombinant colonies were selected by plating in the presence of kanamycin. Gene disruption in the BAC was confirmed by PCR, resulting in a highly efficient process with nearly 100% positive clones. The BAC carrying the disrupted allele(s) was used as a template for a PCR reaction using the Pwo high fidelity DNA polymerase. The final PCR product was subsequently used as AES and contains the Km$^r$ cassette with 1,500 bp homology arms flanking the target gene(s) (using E6C10 P1A/E6C10 P2A or Ag85B P1A/Ag85B P2A primer pairs, Table 1). In parallel, MTBVAC transformed with pJV53H (a kind gift from Christophe Guilhot, IPBS, Toulouse, France), which consists on the replacement of the original Km$^r$ with a hygromycin-resistance marker in pJV53 (ref. 46), was grown in the presence of 0.2% acetamide to allow expression of the recombinase system from this plasmid. The AES was transformed in these cells by electroporation. Recombinant MTBVAC colonies containing the desired mutation were selected by plating on kanamycin and confirmed by PCR amplified with primers flanking the deleted gene (using E6C10 Fw/E6C10 Rv or Ag85B Fw/Ag85B Rv primer pairs, Table 1). Some of the colonies analysed contained both wild-type and mutant amplicons suggesting an unspecific recombination of the AES. We selected colonies where only the mutant amplicon was amplified (Supplementary Fig. 4 and Table 2).

**Table 1 | Primer list.**

| Primer | Primer sequence 5′→3′ | Application |
|---|---|---|
| 16S Fw | ATGACGGCCTTCGGGTTGTAA | qRT–PCR |
| 16S Rv | CGGCTGCTGGCACGTAGTTG | qRT–PCR |
| ESAT6 Fw | AGGGTGTCCAGCAAAAATGG | qRT–PCR |
| ESAT6 Rv | CTGCAGCGCGTTGTTCAG | qRT–PCR |
| CFP10 Fw | GCAGGAGGCAGGTAATTTCG | qRT–PCR |
| CFP10 Rv | CCTGGTCGATCTGGGTTTTC | qRT–PCR |
| Ag85A Fw | ATGCAGCTTGTTGACAGGGTT | qRT–PCR |
| Ag85A Rv | TCGACGCGACATACCCGT | qRT–PCR |
| Ag85B Fw | GGTTCAGTTCCAGAGCGGTG | qRT–PCR |
| Ag85B Rv | TCGAGCAGATAAACCGCAGG | qRT–PCR |
| EspA Fw | GGCACCCTCGGAGAAGTGT | qRT–PCR |
| EspA Rv | AGCTCTTTCAGGCCGTTGAG | qRT–PCR |
| EspC Fw | TGTACTTGACTGCCCACAATGC | qRT–PCR |
| EspC Rv | TCGACACCGGCCGTATG | qRT–PCR |
| E6C10 P1 | ACATTTTGGCGAGGAAGGTAAAGAGAGAAAGTAGTCCAGCGTGTAGGCTGGAGCTGCTTC | MTBVACΔE6C10 |
| E6C10 P2 | GATCCCGTGTTTCGCTATTCTACGCGAACTCGGCGTTGCCCATATGAATATCCTCCTTAGT | MTBVACΔE6C10 |
| E6C10 Fw | TCCCGTAATGACAACAGACTTC | MTBVACΔE6C10 |
| E6C10 Rv | GGAAGAGCTTGTCGTAGTCG | MTBVACΔE6C10 |
| E6C10 P1A | AGCGGATTTGACGTCGTGCT | MTBVACΔE6C10 |
| E6C10 P1B | CGATTGTCGCCCTACCCGAT | MTBVACΔE6C10 |
| Ag85B P1 | CCCGAGCACACGACGACATACAGGACAAAGGGGCACAAGTGTGTAGGCTGGAGCTGCTTC | MTBVACΔAg85B |
| Ag85B P2 | ATCTCACGTGGACGGGTAAGCAACCCTTCGGTTGATCCCGCATATGAATATCCTCCTTAGT | MTBVACΔAg85B |
| Ag85B Fw | ACATTTGGCCTCCACACAC | MTBVACΔAg85B |
| Ag85B Rv | CAATCAGCGACAACAGGATGCC | MTBVACΔAg85B |
| Ag85B P1A | TAGCACTCGAGTGATCGGCT | MTBVACΔAg85B |
| Ag85B P1B | CCTCGAACCAACCGCCTTC | MTBVACΔAg85B |

**Table 2 | Plasmid and BAC information.**

| BAC or plasmids | Source | Reference |
|---|---|---|
| H37Rv BAC Library | Institut Pasteur Paris (France) | 44 |
| pKD46 | Coli Genetic Stock Center (Yale University) | 45 |
| pKD4 | Coli Genetic Stock Center (Yale University) | 45 |
| pJV53H | IPBS (Toulouse, France) | 46 |

**In vitro presentation assay.** Bone marrow cells collected from mouse femurs and tibias were plated on sterile Petri dishes and incubated for 7 days in Dulbecco's modified Eagle's medium (Gibco) containing 10% (v/v) heat-inactivated FCS, 100 units ml$^{-1}$ penicillin/100 mg ml$^{-1}$ streptomycin (Sigma) and 10% (vol/vol) conditioned medium from mouse L929 fibroblasts (a kind gift from Dr Esther Perez, GSK, Tres Cantos, Madrid, Spain). $10^5$ bone-marrow-derived macrophages were seeded in 96 flat-well plates and incubated for 4 h with CFP10 antigen at the indicated concentrations in triplicates. After three washing steps with PBS, $10^6$ purified CD4+ splenocytes obtained by MACS separation (Miltenyi Biotec) (purity higher than 90 %) were added in 200 μl of RPMI 1640 (Sigma) containing 10% (v/v) heat-inactivated FBS, 100 units ml$^{-1}$ penicillin, 100 mg ml$^{-1}$ streptomycin, 50 μM 2-mercaptoethanol (Sigma) and cells were incubated for 24 h. The plate was then centrifuged at 800g, and the supernatant recovered to determine IFNγ by ELISA (MabTECH).

**Protein analysis.** To analyse proteins from extracellular and intracellular compartments, BCG, MTBVAC or H37Rv were cultured in 7H9 liquid medium with 0.05% (v/v) Tween-80 and supplemented with 0.2% (w/v) dextrose and 0.085% (w/v) NaCl, to avoid albumin contamination from ADC in the supernatant fraction. After 3 weeks of incubation, cultures were pelleted by centrifugation and the supernatants were passed through 0.2 μm-pore filters to remove any residual bacteria. Supernatant proteins were precipitated with 10% (v/v) trichloroacetic acid (Sigma) during 1 h in ice and centrifugation at 3,200g during 30 min at 4 °C. Pelleted proteins were rinsed with cold acetone and, after decanting acetone, air dried pellets were resuspended in 150 mM Tris/HCl pH 8. Bacterial pellet was washed twice and resuspended in 1 ml PBS containing a protease inhibitor cocktail (Roche). Bacteria were disrupted by sonication for 30 min at 4 °C using a Bioruptor (Diagenode). Samples were centrifuged 5 min at 18 000g and the supernatant collected for further analysis.

Intracellular and secreted protein concentrations were determined with the QuantiPro BCA Assay kit (Sigma). Ten micrograms of protein per well were loaded and separated by SDS–PAGE. Following protein transference to PVDF membranes, immunodetection was carried out using mouse monoclonal antibodies anti-ESAT6 (1:1,000; clone 11G4; Abcam ref.: ab26246) and anti-GROEL (1:500; clone BDI578; Abcam ref.: ab20045), rabbit polyclonal anti-CFP10 (1:2,000; Thermo Scientific ref.: PA1-19445) and chicken polyclonal anti-Ag85A (1:1,000; Abcam ref.: ab14073). Membranes were incubated with the corresponding HRP-conjugated secondary antibodies (1:20,000; Sigma) and signal developed with the ECL Plus Western Blotting System (GE HealthCare). To reprobe each blot with different antibodies, membranes were incubated with ReBlot Plus Strong Antibody Stripping Solution (Millipore) according to the manufacturer's instructions.

For visualization of proteins following electrophoresis, polyacrylamide gels were stained using a commercial colloidal blue staining kit (Invitrogen) according to the manufacturer's instructions. Bands of interest were split and sent for identification by MALDI-TOF MS to Proteomics Services of the University of Zaragoza.

**qRT–PCR.** Groups of C3H/HeNRj or C57BL/6 8–10 weeks old, female mice were intranasally challenged with $10^3$ CFU of H37Rv in 40 μl of PBS. Four weeks later, mice were killed individually and lungs collected and cut in small pieces, which were homogenized in 4 ml of TRIZOL by mechanical disruption using a Dounce tissue grinder. Lung homogenates were split in four 1-ml aliquots and placed on dry ice until further processing. Then, 200 μl of chloroform were added per ml of TRIZOL and after vigorous vortexing, tubes were centrifuged at 18,000g during 1 h at 4 °C. Aqueous upper phase containing eukaryotic RNA was recovered and stored at −80 °C. Remaining supernatant was discarded and the four pellets containing mycobacteria were resuspended and collected with one additional millilitre of TRIZOL and 200 μl of chloroform. Bacteria were disrupted adding glass beads using a Fast Prep device, with two 45″ cycles at highest speed. Then, tubes were centrifuged 10 min at 18,000g (4 °C) and aqueous phase (500 μl approx.) containing mycobacterial RNA (and also remaining eukaryotic RNA) were recovered. A measure of 700 μl of isopropanol was added and tubes were incubated at room temperature during 15 min to favour RNA precipitation. Precipitated nucleic acids were collected by centrifugation. The pellets were rinsed with 70% ethanol and air dried before being re-dissolved in RNase-free water. DNA was removed from RNA samples with TurboDNAfree (Ambion) by incubation at 37 °C for 1 h. RNA integrity was assessed by agarose gel electrophoresis, and absence of contaminating DNA was checked by lack of amplification products after 30 PCR cycles. Lungs from uninfected animals were processed as above to be used as non-infected control in the qPCR.

For qRT–PCR, retro-transcription of 1 μg of RNA was performed following a standard reverse transcription reaction with the SuperScript Reverse Transcriptase (Invitrogen). The generated cDNA served as a template for qPCR in the presence of gene-specific primers (Table 1) and SYBR Green Master-Mix (Roche). Four

replicates of each gene $C_T$ value were obtained in the StepOne Plus Instrument (Applied Biosystems) and were normalized to the $C_T$ of the *16s* rRNA gene (amplified from the same samples), obtaining a $\Delta C_T = C_{T,j} - C_{T,16s}$, where $j$ is a gene different from *16s*. Our observations indicated unspecific gene amplification in the non-infected cDNA controls. As a result, following a procedure previously reported[47], we calculated a $\Delta\Delta C_T$ value specific for infection ($\Delta\Delta C_T(\text{inf})$), subtracting the $\Delta C_T$ mean value for each gene obtained from non-infected controls from the $\Delta C_T$ obtained for each sample from infected mice. Finally, change was calculated with the equation $2^{-\Delta\Delta C_T(\text{inf})}$. Gene expression analysis from *in vitro* samples was performed using conventional comparative $C_T$ method. Normalized gene expression data (using the *16s* rRNA as housekeeping gene) were represented as the fold-change compared to *fbpB* expression for each of the *in vivo* and *in vitro* conditions tested. The melting curves for all gene-specific primer pairs (sequences used in Supplementary Experimental Procedures) were examined to identify primer-dimer formation and to ensure the uniformity of the amplicons.

**Statistical analysis.** Results from this study were not blinded for analysis. No statistical method was used to calculate sample size in animal experiments. GraphPrism software was used for statistical analysis. Brown–Forsythe test was used to assess variances homogeneity. Variances were similar among compared groups. Statistical tests used for each experiment are indicated in the figure legends. All statistical tests used were two-tailed. Outlier values were determined applying the Grubb's test to all data sets, and were discarded from the final statistical analysis. Differences were considered significant at $P < 0.05$.

**Data availability.** The authors declare that the data supporting the findings of this study are available in this article and its Supplementary Information Files, or from the corresponding authors on request.

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

## Acknowledgements

The authors acknowledge the Scientific and Technical Services from Instituto Aragonés de Ciencias de la Salud and Universidad de Zaragoza. This work was supported by the Spanish Ministry of Economy and Competitiveness (Grant number BIO2014-5258P), the European Commission H2020 program (Grant number TBVAC2020 643381) and 'Gobierno de Aragón/Fondo Social Europeo'.

## Author contributions

N.A., F.S and C.M. designed the experiments and directed the study. N.A., J.G.-A., S.A.A., A.B.G., S.U. and R.A. performed the experiments. R.S. and M.S. provided the ESAT6 and CFP10 recombinant antigens. N.A., D.M. and C.M. wrote the manuscript. The funders had no role in study design, data collection and analysis, decision to publish or preparation of the manuscript.

## Additional information

**Competing interests:** C.M. and J.G.-A. are co-inventors in a patent application entitled "Tuberculosis vaccine" filled by the University of Zaragoza (application number: PCT/ES 2007/070051). The remaining authors declare no competing financial interests.

