## [Peer Review File · Nature Communications]

Reviewers' comments:

Reviewer #1 (Remarks to the Author):

These authors have studied the immunogenicity and protective efficacy of a viable, attenuated vaccine (MTBVAC) in three strains of inbred mice. They claim that protection correlates with the immunological response to two antigens, CFP10 and ESAT-6, which are not found in BCG. They also claim that MTBVAC protects better than BCG.

The manuscript is diffuse and unfocused. Most of the data in the Supplemental Data section are merely duplicative of previously published results (Supplementary Figures 1 & 3) or add nothing to the main point of the paper (Supplementary Figure 4) and can be eliminated.

Virtually all of the data that support the authors' conclusions were obtained in a single strain of mice, namely, C3H. There is no evidence in two commonly used mouse strains (C57BL/6, BALB/c) that MTBVAC protects better than BCG. Furthermore, the authors do not demonstrate a "correlation" between reactogenicity to CFP-10 and ESAT-6 and protection even in C3H mice. The term "correlation" has a very specific statistical meaning and the authors did not fulfill the analytical requirements that allow them to use this term.

The questions which the authors should be addressing are: (1) What is weird about C3H mice?; (2) Why should we expect humans to behave like C3H mice instead of C57BL/6 or BALB/c mice?; (3) What is the biological relevance of the results obtained in this singular mouse strain?

The authors repeatedly imply that it is the secretion of the antigens from MTBVAC that is critical for the immunological responses and protection. However, they offer no evidence for this statement. There is plenty of intracellular antigen and no reason to expect that organisms which are killed within host phagocytic cells would not provide plenty of antigen to activate T cells.

The authors claim that that CD4+ T cells are the "major contributors" of IFN γ (Supplemental Figure 2). However, the authors do not provide any data on other likely sources of IFN γ , e.g., CD8+ T cells.

The authors repeatedly imply that it is the unique MHC haplotype of C3H mice that explains the different biological behavior of MTBVAC in that strain. However, the authors have not eliminated the obvious alternative hypotheses that might contribute equally well to their observations. For example, MTBVAC might replicate or disseminate differently than BCG in C3H mice, both of which would have a significant impact on protection. The authors provide no data on the biological behavior (e.g., growth, dissemination) of MTBVAC vs BCG in their mice.

The authors infected their mice with virulent *M. tuberculosis* by the intranasal route. A much better route would have been by aerosol. In the Methods section, the authors state that animals were infected with a specific dose (e.g., 10-20 CFU, 150 CFU, 1000 CFU, etc) for different experiments. Please provide the evidence that the number of viable mycobacteria specified in the Methods was actually reached the lung immediately following intranasal instillation.

Reviewer #2 (Remarks to the Author):

This manuscript by Aguilo et al presents the results of a vaccination study using MTBVAC, a new vaccine candidate in clinical development in comparison to BCG. The results shown in the manuscript reveal very interesting insights into antigenicity and protective efficacy of proteins that

are present in the MTBVAC vaccine but are absent from BCG. The authors describe that in different mouse models, significant better protection can be observed in the C3H/HeNRj model with MTBCAV relative to BCG because in these mice, antigen specific interferon-gamma responses against CFP10 are obtained.

Overall the paper describes important and interesting observations. However, there are also some points for which the manuscript could be improved. Please see below the different questions and suggestions.

Line (L) 50 : The sentence : "This region is located in the ESX-1 secretion system " needs revision ESX-1-encoding region

L 66: sentence "... virulence lipids constituent of the lipid capsule 12. " is confusing as the capsule is normally not built out of lipids.

L 131 vaccinated group as compared to the group infected with H37Rv (Supplementary Fig. 3d). In line with these observations, the Phase 1 clinical data showed that.

The authors present this situation, as remarkably, however this result was to be expected as in MTBVAC ESAT-6 is not secreted due to the interruption of the PhoP-EspR-EspACD-ESXA co-secretion regulation.

Additionally, an MTBVAC knockout for Ag85B was constructed to evaluate the role of this antigen in MTBVAC-conferred protection. Results showed that in the absence of Ag85B-specific response MTBVAC efficacy remained unaffected in any of the three genetic backgrounds tested (Fig. 3).

This is interesting data, but how do the authors interpret this result in the light of previous better protection conferred by an rBCG strain overexpressing Ag85B (Horwitz). Furthermore, the authors have previously shown that MTBVAC over-secretes Ag85b due to a PhoP regulated interfering small RNA. How can the previous data and the currently obtained data be comprehensively combined?

I suggest to the authors that they discuss their results also in the light of their previous papers where they show that small RNA is regulating the secretion of AG85B. How do the authors compare these results to the AG85B overexpressing BCG strain that showed better protection?

by MTBVAC. Indeed, analysis of the bacterial production and secretion of ESAT6 and CFP10 in MTBVAC revealed that, whereas ESAT6 was found only intracellularly, CFP10 was also present in the secreted fraction of MTBVAC (Supplementary Fig. 7). In this regard, the differential ability of bacterial secreted antigens to interact with the immune system with regards to intracellular proteins has been previously described by other authors, suggesting that this phenomenon could have an impact in protective efficacy.

This observation is very interesting although it challenges the current type VII secretion model. To make the point by the authors stronger, I would suggest to move the supplementary Figure S5 to the main manuscript.

In this relation the authors might also discuss their vision how such a separated secretion would be possible.

It would be interesting to express CFP-10 alone in the ESAT-6/CFP-10 deleted MTBVAC strain.

L299 protection. In two other works, 8,28 where we vaccinated C57BL/6 adult mice, we "works" might be exchanged with "studies"

L 210 ESAT6 with CFP10) 21,22 relative to fbpA and fbpB. There has been a new paper published on EspC forming a putative secretion needle (Lou et al., 2016). This information might be included and discussed.

L650 Figure 2. Immune response specific to CFP10 correlates with the improved
651 protection conferred by MTBVAC. (a) Immunoblot analysis of GroEL2, ESAT6 and
652 CFP10 in MTBVAC and MTBVAC γ E6C10 lysate samples. (b, c)

It would be interesting to see also the immunoblot analyses for the culture filtrates of these strains.

Point-by-point response to reviewers' comments

Reviewer #1

1) Reviewer's comment: The manuscript is diffuse and unfocused. Most of the data in the Supplemental Data section are merely duplicative of previously published results (Supplementary Figures 1 & 3) or add nothing to the main point of the paper (Supplementary Figure 4) and can be eliminated.

Author's response: In the revised version of the manuscript, we have tried to address all the concerns of the reviewer to transmit a more concrete and focused message.

Regarding supplementary information, we have modified this section following reviewer's advise. Supplementary Figure 1 now includes a dot-plot diagram with the distribution of CD4+ and CD8+ cells that express IFN γ , as suggested by reviewer (see below our response to comment 5). Supplementary Figure 2 (previous Supplementary Figure 1), maintains the SDS-PAGE image showing the differential secretion of Ag85B by MTBVAC as we consider it important for interpreting some of the immunogenicity data of the study, and these results had not been previously published. We agree that Supplementary Figures 3 & 4 are not relevant for the conclusions of the study, and therefore we have eliminated them from the revised manuscript. Following reviewer's comments (see response to comments 6 and 7, respectively, below), we have included data of new experiments in two new figures: Supplementary Figure 3 showing vaccine replication and dissemination of MTBVAC and BCG in C3H mice, and Supplementary Figure 6 including data of the initial lung bacterial load following H37Rv intranasal challenge.

2) Reviewer's comment: "Virtually all of the data that support the authors' conclusions were obtained in a single strain of mice, namely, C3H. There is no evidence in two commonly used mouse strains (C57BL/6, BALB/c) that MTBVAC protects better than BCG. Furthermore, the authors do not demonstrate a "correlation" between reactogenicity to CFP-10 and ESAT-6 and protection even in C3H mice. The term "correlation" has a very specific statistical meaning and the authors did not fulfill the analytical requirements that allow them to use this term".

Author's response: We apologize if we had not transmitted clearly our premise in the

manuscript. We consider that our conclusions are based on the findings obtained in the three mouse strains tested, not only in the C3H. Our initial hypothesis was based on the search for variability in BCG- and MTBVAC-induced protection and the differential antigen-specific response against ESAT6 and CFP10 between the three mouse strains tested, with the objective to identify a potential role in protection of the specific immunogenicity against these antigens. Our findings exemplify how host genetic background could affect antigen immunodominance and vaccine-induced protection against tuberculosis. Considering the great genetic variability among human populations, it results crucial to identify host-specific biomarkers that could help anticipate which individuals may show improved response to vaccine-induced protection and which may not in clinic.

We agree with the reviewer that the term “correlation” is not appropriate and therefore we have limited its use in the revised version of the manuscript. We have changed the title to address this question, replacing the term “correlates with” by “linked to”.

3) Reviewer’s comment: The questions which the authors should be addressing are:

(1) What is weird about C3H mice?

Author’s response: C3H is an inbred strain whose haplotype (H-2k) is different from that of C57BL/6 (H-2b) and BALB/c (H-2d) and therefore, shows a differential capacity to recognize and present peptides. Our reasoning to include the C3H mice in our studies was inspired by a previous work of Samuel Behar ¹, where they described the differential ability of C3H mice (compared to C57BL/6 or BALB/c) to recognize the major tuberculosis antigen CFP10. As the two major TB antigens CFP10 and ESAT6 are differentially expressed by MTBVAC in comparison to BCG, we hypothesized that a host able to react against these two proteins following MTBVAC vaccination could confer a better protection than BCG.

We agree with the reviewer that this question was not well addressed in the manuscript and as such, we have tried to better explain it in the new revised version, at the end of the introduction (lines 82-90), and results (lines 117-121).

(2) Why should we expect humans to behave like C3H mice instead of C57BL/6 or BALB/c mice?

Author’s response: Our rationale for anticipating that our results could have an impact in

clinic comes from the fact that up to 80% of human haplotypes recognize both ESAT6 and CFP10 ^{2,3} (similar to C3H). In this regard, our preclinical data suggest that MTBVAC might confer improved protection in the human population reactogenic to ESAT6 & CFP10. Conversely, in the remaining human population, whose haplotypes do not recognize derived peptides from these two antigens, our results may indicate that MTBVAC could not induce improved efficacy relative to BCG. However, we are aware of the importance of testing all these data in MTBVAC clinical trials.

We have now tried to make this argument stronger in the discussion section of the new manuscript (lines 315-326).

(3) What is the biological relevance of the results obtained in this singular mouse strain?"

Author's response: ESAT6 and CFP10 are the two most antigenic tuberculosis proteins of special interest in the tuberculosis vaccine development field. Our data indicate that deletion of these two antigens in MTBVAC abrogates its improved protection over BCG in C3H mice but not in C57BL/6 and BALB/c (Figure 3). Considering the comparable biological behaviour (dissemination and replication capacity) of MTBVAC and BCG (see response to comment 6 below) in the C3H mice, our results suggest that the differential immune response specific for CFP10 and ESAT6 elicited by MTBVAC contributes to improved protection relative to BCG. As MTBVAC is the only vaccine in the clinical pipeline to date ⁴ able to express both antigens, confirming these results in next MTBVAC human evaluation trials could greatly accelerate its clinical development.

We consider that this idea is now better-reflected in the manuscript.

4) Reviewer's comment: The authors repeatedly imply that it is the secretion of the antigens from MTBVAC that is critical for the immunological responses and protection. However, they offer no evidence for this statement. There is plenty of intracellular antigen and no reason to expect that organisms which are killed within host phagocytic cells would not provide plenty of antigen to activate T cells.

Author's response: We agree with the reviewer that intracellular antigens can activate T cells. Indeed, our results show that ESAT6, which is expressed intracellularly but not secreted by MTBVAC, is immunogenic upon vaccination with MTBVAC. In this regard, we agree that the statement indicated by the reviewer is too speculative for inclusion in the results section, and as such we have only maintained it in the discussion.

In line with another reviewer's suggestion, in Figure 1 of the revised version of the manuscript, we have now included results on the differential expression and secretion levels of CFP10 and ESAT6 by MTBVAC relative to BCG, without suggesting in the results section any potential implication of this data in the generation of a differential immune response.

5) Reviewer's comment: The authors claim that that CD4+ T cells are the "major contributors" of IFN γ (Supplemental Figure 2). However, the authors do not provide any data on other likely sources of IFN γ , e.g., CD8+ T cells.

Author's response: We agree with the reviewer in this point, and in the revised manuscript we provide new data (including IFN γ + CD8+ T cell results) in the now Supplementary Figure 1 to support this statement. We have included these results in the results section (lines 122-127).

6) Reviewer's comment: The authors repeatedly imply that it is the unique MHC haplotype of C3H mice that explains the different biological behaviour of MTBVAC in that strain. However, the authors have not eliminated the obvious alternative hypotheses that might contribute equally well to their observations. For example, MTBVAC might replicate or disseminate differently than BCG in C3H mice, both of which would have a significant impact on protection. The authors provide no data on the biological behavior (e.g., growth, dissemination) of MTBVAC vs BCG in their mice.

Author's response: We thank the reviewer for this comment and accordingly, we have performed an additional experiment to evaluate BCG and MTBVAC replication and dissemination capacity in draining lymph nodes and spleen (main lymphoid organs) of C3H mice at 1, 2 and 4 weeks post-vaccination. These data are now included in the results section of the revised manuscript (lines 145-151). This experiment is similar to a previous one conducted in the BALB/c mouse strain, where the biodistribution profile of MTBVAC and BCG was shown to be highly comparable, in agreement with Regulatory safety requirements to support entry of MTBVAC into first-in-human clinical evaluation ⁵.

7) Reviewer's comment: The authors infected their mice with virulent M. tuberculosis by the intranasal route. A much better route would have been by aerosol. In the Methods section, the authors state that animals were infected with a

specific dose (e.g., 10-20 CFU, 150 CFU, 1000 CFU, etc) for different experiments. Please provide the evidence that the number of viable mycobacteria specified in the Methods was actually reached the lung immediately following intranasal instillation.

Author's response: In addition to aerosol delivery, the intranasal route is also commonly used for pulmonary administration in tuberculosis challenge experiments ^{6,7}, suggesting that both routes are acceptable.

We have included the requested data by the reviewer to provide evidence that number of viable mycobacteria specified in the Methods section was actually reached in the lungs immediately following intranasal instillation. In the present version of the revised manuscript, we have included a new Supplementary Figure 6 with these results.

Reviewer #2

1) Reviewer's comment: Line (L) 50: The sentence "This region is located in the ESX-1 secretion system" needs revision ESX-1-encoding region.

Author's response: corrected in the revised version.

2) Reviewer's comment: L 66: sentence ..." virulence lipids constituent of the lipid capsule 12. " is confusing as the capsule is normally not built out of lipids.

Author's response: corrected in the revised version.

3) Reviewer's comment: L131 vaccinated group as compared to the group infected with H37Rv (Supplementary Fig. 3d). In line with these observations, the Phase 1 clinical data showed that.

The authors present this situation, as remarkably, however this result was to be expected as in MTBVAC ESAT-6 is not secreted due to the interruption of the PhoP-EspR-EspACD-ESXA co-secretion regulation.

Author's response: We agree with the reviewer that the result obtained in that experiment was the expected and therefore the situation should not have been presented as remarkable. In the new version of the revised manuscript, this supplementary figure and this part of the text have been removed following the advise of another reviewer.

4) Reviewer's comment: "Additionally, an MTBVAC knockout for Ag85B was constructed to evaluate the role. of this antigen in MTBVAC-conferred protection. Results showed that in the ; absence of Ag85B-specific response MTBVAC efficacy remained unaffected in any of the three genetic backgrounds tested (Fig. 3)".

This is interesting data, but how do the authors interpret this result in the light of previous better protection conferred by an rBCG strain overexpressing Ag85B (Horwitz). Furthermore, the authors have previously shown that MTBVAC over-secretes Ag85b due to a PhoP regulated interfering small RNA. How can the previous data and the currently obtained data be comprehensively combined? I suggest to the authors that they discuss their results also in the light of their previous papers where they show that small RNA is regulating the secretion of

AG85B. How do the authors compare these results to the AG85B overexpressing BCG strain that showed better protection?

Author's response: In agreement with the reviewer about the relevance of discussing our data with the results from the Ag85B-overexpressing BCG vaccine, rBCG30, we have included a paragraph in the discussion section of the revised manuscript version (lines 252-257).

We thank the reviewer for the comment about PhoP-regulated interfering small RNA that regulates Ag85B secretion ⁸. We have included this reference in the current version.

5) Reviewer's comment: "by MTBVAC. Indeed, analysis of the bacterial production and secretion of ESAT6 and CFP10 in MTBVAC revealed that, whereas ESAT6 was found only intracellularly, CFP10 was also present in the secreted fraction of MTBVAC (Supplementary Fig. 7). In this regard, the differential ability of bacterial secreted antigens to interact with the immune system with regards to intracellular proteins has been previously described by other authors, suggesting that this phenomenon could have an impact in protective efficacy".

This observation is very interesting although it challenges the current type VII secretion model. To make the point by the authors stronger, I would suggest to move the supplementary Figure S5 to the main manuscript. In this relation the authors might also discuss their vision how such a separated secretion would be possible. It would be interesting to express CFP-10 alone in the ESAT-6/CFP-10 deleted MTBVAC strain.

Author's response: We agree with the reviewer that ESAT6-independent secretion of CFP10 by MTBVAC is interesting and unexpected as it challenges in some way the current model. Thus, following reviewer's suggestion to make this point stronger we have moved the Supplementary Figure 5 to the main body of the revised manuscript. The new Figure 1 displays the results of the expression and secretion levels of ESAT6 and CFP10 by MTBVAC (compared to BCG and *M. tuberculosis*). We have included an additional paragraph to discuss these observations in the discussion section (lines 304-314).

We agree that further studies, as those suggested by the reviewer, are needed to understand the molecular mechanisms leading to CFP10 secretion uncoupled to ESAT6 and they will be of interest in the future, but we consider that molecular comprehension of this phenomenon is out of the scope of this study. However, we have reflected in the

discussion section the interest of these experiments (lines 301-303).

6) L299 protection. In two other works 8,28 where we vaccinated C57BL/6 adult mice, we “works” might be exchanged with “studies”.

Author’s response: corrected in the revised version of the manuscript.

7) Reviewer comment’s: L 210 ESAT6 with CFP10) 21,22 relative to fbpA and fbpB. There has been a new paper published on EspC forming a putative secretion needle (Lou et al., 2016). This information might be included and discussed.

Author’s response: This citation has now been incorporated in the new version of the revised manuscript.

8) Reviewer’s comment: L650 Figure 2. Immune response specific to CFP10 correlates with the improved protection conferred by MTBVAC. (a) Immunoblot analysis of GroEL2, ESAT6 and CFP10 in MTBVAC and MTBVAC γ E6C10 lysate samples. (b, c). It would be interesting to see also the immunoblot analyses for the culture filtrates of these strains.

Author’s response: As the objective of the immunoblot shown in the original Figure 2 (current Figure 3 of their revised manuscript) was to confirm the absence of ESAT6 and CFP10 expression in the mutant strain generated, we only included the intracellular fraction analysis. In addition lack of ESAT6 and CFP10 expression is confirmed by the lack of antigen-specific immunogenicity by the MTBVAC mutant strain against ESAT6/CFP10 (current Figure 3).

REFERENCES

- 1 Kamath, A. B. *et al.* Cytolytic CD8+ T cells recognizing CFP10 are recruited to the lung after Mycobacterium tuberculosis infection. *J Exp Med* **200**, 1479-1489, doi:10.1084/jem.20041690 (2004).
- 2 Lindestam Arlehamn, C. S. *et al.* A Quantitative Analysis of Complexity of Human Pathogen-Specific CD4 T Cell Responses in Healthy M. tuberculosis Infected South Africans. *PLoS Pathog* **12**, e1005760, doi:10.1371/journal.ppat.1005760 (2016).
- 3 Millington, K. A. *et al.* Rv3615c is a highly immunodominant RD1 (Region of Difference 1)-dependent secreted antigen specific for Mycobacterium tuberculosis infection. *Proceedings of the National Academy of Sciences of the United States of America* **108**, 5730-5735, doi:10.1073/pnas.1015153108 (2011).
- 4 WHO Global Tuberculosis Report 2016. <http://apps.who.int/iris/bitstream/10665/250441/1/9789241565394-eng.pdf?ua=1>.
- 5 Arbues, A. *et al.* Construction, characterization and preclinical evaluation of MTBVAC, the first live-attenuated M. tuberculosis-based vaccine to enter clinical trials. *Vaccine* **31**, 4867-4873, doi:10.1016/j.vaccine.2013.07.051 (2013).
- 6 Liu, Y. *et al.* Immune activation of the host cell induces drug tolerance in Mycobacterium tuberculosis both in vitro and in vivo. *J Exp Med* **213**, 809-825, doi:10.1084/jem.20151248 (2016).
- 7 Wang, J. *et al.* Single mucosal, but not parenteral, immunization with recombinant adenoviral-based vaccine provides potent protection from pulmonary tuberculosis. *J Immunol* **173**, 6357-6365 (2004).
- 8 Solans, L. *et al.* The PhoP-dependent ncRNA Mcr7 modulates the TAT secretion system in Mycobacterium tuberculosis. *PLoS Pathog* **10**, e1004183, doi:10.1371/journal.ppat.1004183 (2014).

REVIEWERS' COMMENTS:

Reviewer #1 (Remarks to the Author):

The authors have responded adequately to my previous criticisms.

Reviewer #2 (Remarks to the Author):

The authors have revised their manuscript in line with the comments of the reviewers. In particular they have better explained the rationale why C3H mice were used and how the immune reactions of C3H mice are linked with the reaction of a very large percentage of humans. The authors have also transferred some of the data from the supplementary material to the main figures, as suggested. In this way it is clearer for the reader to appreciate the CFP-10 Western blots. Moreover, the authors have updated the references and included some of the previously lacking information.

In conclusion, in my opinion the authors have successfully improved the manuscript by carefully taking into account the various referees' comments.

Point-by-point response to reviewers' comments

We thank the reviewers for their positive comments and we are glad to observe that reviewers are satisfied with our responses to their concerns.

REVIEWERS' COMMENTS:

Reviewer #1 (Remarks to the Author):

The authors have responded adequately to my previous criticisms.

Reviewer #2 (Remarks to the Author):

The authors have revised their manuscript in line with the comments of the reviewers. In particular they have better explained the rationale why C3H mice were used and how the immune reactions of C3H mice are linked with the reaction of a very large percentage of humans. The authors have also transferred some of the data from the supplementary material to the main figures, as suggested. In this way it is clearer for the reader to appreciate the CFP-10 Western blots. Moreover, the authors have updated the references and included some of the previously lacking information. In conclusion, in my opinion the authors have successfully improved the manuscript by carefully taking into account the various referees' comments.